# Optimization of Thermoelectric Properties and Physical Mechanisms of Cu_2_Se-Based Thin Films via Heat Treatment

**DOI:** 10.3390/nano14171421

**Published:** 2024-08-30

**Authors:** Haobin Li, Fu Li, Yuexing Chen, Guangxing Liang, Jingting Luo, Meng Wei, Zhi Zheng, Zhuanghao Zheng

**Affiliations:** 1Shenzhen Key Laboratory of Advanced Thin Films and Applications, Key Laboratory of Optoelectronic Devices and Systems of Ministry of Education and Guangdong Province, State Key Laboratory of Radio Frequency Heterogeneous Integration, College of Physics and Optoelectronic Engineering, Shenzhen University, Shenzhen 518060, China; 2210452032@email.szu.edu.cn (H.L.); lifu@szu.edu.cn (F.L.); chenyx@szu.edu.cn (Y.C.); lgx@szu.edu.cn (G.L.); luojt@szu.edu.cn (J.L.); zhengzh@szu.edu.cn; 2Key Laboratory for Micro-Nano Materials for Energy Storage and Conversion of Henan Province, Institute of Surface Micro and Nano Materials, College of Chemical and Materials Engineering, Xuchang University, Xuchang 461000, China; zzheng@xcu.edu.cn (Z.Z.)

**Keywords:** thermoelectric thin films, Cu_2_Se, in situ growth, thermoelectric properties

## Abstract

Cu_2_Se is an attractive thermoelectric material due to its layered structure, low cost, environmental compatibility, and non-toxicity. These traits make it a promising replacement for conventional thermoelectric materials in large-scale applications. This study focuses on preparing Cu_2_Se flexible thin films through in situ magnetron sputtering technology while carefully optimizing key preparation parameters, and explores the physical mechanism of thermoelectric property enhancement, especially the power factor. The films are deposited onto flexible polyimide substrates. Experimental findings demonstrate that films grown at a base temperature of 200 °C exhibit favorable performance. Furthermore, annealing heat treatment effectively regulates the Cu element content in the film samples, which reduces carrier concentration and enhances the Seebeck coefficient, ultimately improving the power factor of the materials. Compared to the unannealed samples, the sample annealed at 300 °C exhibited a significant increase in room temperature Seebeck coefficient, rising from 9.13 μVK^−1^ to 26.73 μVK^−1^. Concurrently, the power factor improved from 0.33 μWcm^−1^K^−2^ to 1.43 μWcm^−1^K^−2^.

## 1. Introduction

Thermoelectric technology has the capability to convert thermal and electrical energy, making it suitable for utilization in portable wearable devices that can convert the temperature difference between the human body and the external environment into electrical energy [1,2,3,4,5,6,7,8]. The efficiency of thermoelectric materials can be assessed by the figure of merit *ZT* (*ZT* = *S*^2^*σT*/*κ*, where *S*, *σ*, *κ*, and *T* represent the Seebeck coefficient, electrical conductivity, total thermal conductivity, and absolute temperature, respectively) [9,10,11]. Due to the challenges associated with measuring *κ* in thin films, the power factor *PF* = *S*^2^*σ* is commonly employed to evaluate thermoelectric properties. It is evident that a high Seebeck coefficient, high electrical conductivity, and low thermal conductivity are essential characteristics for achieving efficient thermoelectric conversion [12,13,14,15]. Currently, most thermoelectric films are fabricated using Bi-Te- and Sb-Te-based materials, which exhibit high thermoelectric properties at room temperature. However, due to their toxicity and high cost implications, they pose difficulties for industrial-scale production [16,17]. Therefore, there is a need to identify an environmentally friendly material with low toxicity and cost-effectiveness that possesses a high *ZT* value for preparing thermoelectric thin films.

Due to its non-toxic nature as well as abundant availability, Cu_2_Se is considered to be a candidate material capable of replacing conventional Te-based materials, thereby offering significant development prospects. Pure bulk Cu_2_Se exhibits a maximum *ZT* value of 1.8, while Yang et al. [18] reported bulk Cu_2_Se having a remarkable *ZT* value reaching up to 2.5. Lei et al. [19] incorporated SiC nanoparticles into the Cu_2_Se matrix, resulting in additional strain and stress within the films. This modification facilitated enhanced migration of charged copper ions, leading to a sevenfold improvement in electrical conductivity compared to pristine Cu_2_Se. As a result, the modified films achieved a peak dimensionless thermoelectric figure of merit of 2.0 at 875 K. Zhao et al. [20] fabricated a range of samples with varying copper content using chemo- and spark-plasma sintering methods. Their investigation revealed that an excess of copper content could diminish the concentration of copper vacancies and sequentially reduce the carrier concentration. Moreover, the surplus copper facilitated the formation of nanopores within the films, effectively decreasing their lattice thermal conductivity. As a result, the dimensionless thermoelectric figure of merit for their samples reached 1.44 at 873 K. The breakthroughs in bulk research have also garnered significant attention towards flexible thin films of Cu_2_Se.

Wang et al. [21] employing pulsed laser deposition (PLD) techniques, synthesized Cu_2_Se thin films, and optimized film characteristics by adjusting the deposition temperature and stoichiometric ratios. These modifications resulted in achieving a power factor of 8.44 μWcm^−1^K^−2^ at ambient temperature, alongside a dimensionless thermoelectric figure of merit (*ZT*) of approximately equal to 0.58 at 580 K. Scimeca et al. [22], through immersion in a Cu+ ion solution, effectively manipulated carrier concentration within Cu_2_Se thin films ranging from 4.3 × 10^21^ to 3.4 × 10^20^ cm^−3^. This strategic adjustment led to a substantial enhancement in the power factor, increasing it by 200–300%. Jin et al. [23] synthesized C-60/Cu_2_Se thermoelectric nanocomposites by integrating varying concentrations of fullerenes into Cu_2_Se. This approach notably augmented the Seebeck coefficient while reducing the lattice thermal conductivity. Consequently, these modifications led to an about 20–30% improvement in dimensionless thermoelectric figure of merit. Zheng et al. [17] enhanced the Cu content in Cu_2_Se films via co-sputtering, utilizing dual targets of Cu_2_Se and Cu. This modification effectively reduced the carrier concentration, increased the Seebeck coefficient, and optimized the electrical and thermal transport properties. Consequently, these adjustments facilitated the achievement of a dimensionless thermoelectric figure of merit of up to 0.42.

Magnetron sputtering is a highly efficient vacuum physical vapor deposition technique that enables control over the component ratio in the samples and is suitable for large-scale industrial preparation [24,25]. In addition, this technology operates under a very high vacuum during the preparation process, effectively preventing sample oxidation [26]. In this study, Cu_2_Se films were prepared on flexible substrates using magnetron sputtering with appropriate base temperature selection and heat treatment to enhance film crystallinity and regulate Cu composition, resulting in an increased Seebeck coefficient and improved *PF* value. The films fabricated at an initial temperature of 200 °C exhibited enhanced thermoelectric properties following a 300 °C annealing process. Cu_2_Se exhibits an α-phase at room temperature and transitions to a β-phase at elevated temperatures, with the phase transition occurring at approximately 400 K. The deposition and annealing temperatures employed in this experiment surpassed this threshold, thus ensuring that the synthesized samples were of the β-phase Cu_2_Se, characterized by a body-centered cubic structure. Specifically, at room temperature, the Seebeck coefficient increased to 26.73 μVK^−1^, and the power factor improved to 1.43 μWcm^−1^K^−2^. Furthermore, when evaluated at a testing temperature of 300 °C, the Seebeck coefficient of the samples further escalated to 43.83 μVK^−1^, while the power factor achieved a peak value of 2.3 μWcm^−1^K^−2^.

## 2. Materials and Methods

Cu_2_Se films were synthesized using a Cu_2_Se alloy target (99%). A flexible polyimide (PI) substrate with a thickness of 0.125 mm and 70% transmittance was employed. The substrate was cleaned sequentially in ethanol, acetone, and ultrapure water, and was sequentially dried using a compressed air jet device and a drying oven. Prior to film deposition, the PI substrate was preheated to 150 °C, 200 °C, and 250 °C to investigate the influence of growth temperature on the thermoelectric properties of the films. The power of the Cu_2_Se target was maintained at 40 W, with a background pressure to 8 × 10^−4^ Pa, an operating pressure of 0.5 Pa, and a sputtering deposition time of 60 min. Once the samples had cooled to room temperature within the vacuum chamber, they were subsequently transferred to a hotplate within a glove box. Sequentially, the samples were annealed for 30 min at 200 °C, 250 °C, and 300 °C in a glove box heating system to evaluate the effect of annealing temperature on the thermoelectric properties of the films.

Measurements of electrical conductivity and the Seebeck coefficient were performed using a thermoelectric property test system (SBA458, Netzsch, Netzsch, Selb, Germany). The carrier concentration and mobility were performed using a Hall measurement system (HL5500PC, Nanometrics, Kanata, ON, Canada). The estimated errors limited by equipment specifications of Seebeck coefficient, electrical conductivity, carrier concentration and mobility are 7%, 7%, 10%, and 10%, respectively. The crystal structure of the films was characterized using an X-ray diffractometer (XRD. D/max 2500, Rigaku Corporation, Tokyo, Japan, using CuKα radiation) scanning from 10° to 70° at a rate of 2° per minute. The surface morphology and chemical composition of the films were examined using scanning electron microscopy (SEM, Zeiss-spra 55, Oberkochen, Baden-Württemberg, Germany) and energy dispersive spectrometry (SEM-EDS, Bruker Quantax 200, Billerica, MA, USA).

## 3. Results and Discussion

Table 1 presents the composition and thickness variations of Cu_2_Se films deposited at different substrate temperatures. An inverse relationship between deposition temperature and film thickness was observed, with its thickness reducing from 322 nm at 150 °C to 276 nm at 250 °C. The Cu: Se stoichiometry exceeded the stoichiometric 2:1 ratio inherent to Cu_2_Se, escalating from 2.07:1 at 150 °C to 2.98:1 at 250 °C, with rising deposition temperatures. This increase in the Cu: Se ratio may be ascribed to the elevated saturated vapor pressure at increased temperatures, which promotes the preferential reaction of selenium due to its higher vapor pressure. The XRD patterns of the thin film samples at different deposition temperatures are shown in Figure 1a. The principal diffraction peaks are discernible at 26.76° and 44.41°, corresponding to the (111) and (220) crystallographic planes of Cu_2_Se, respectively, in agreement with the standard reference pattern (PDF#71-0044). The crystalline grains predominantly grow along the (111) crystal plane at 150 °C, rather than the (220) or (331) planes. As the deposition temperature increases, a noticeable shift in the growth orientation of the crystalline grains occurs, with enhanced prominence along the (220) and (331) planes. Figure 1b–d displays a comparative analysis of the surface morphology of the film samples across a range of deposition temperatures. With an increase in the substrate temperature, a slight enhancement in the surface grain size of the film is observed, accompanied by densification and a limited number of columnar particles. The morphology of films deposited at 150 °C resembles that of samples deposited at 200 °C, exhibiting no significant disparities in densification. However, the XRD pattern lacks (220) and (311) peaks due to inadequate crystallinity of the sample. Films deposited at 250 °C exhibit columnar clusters on their surfaces, potentially influencing their thermoelectric properties.

The thermoelectric properties of the films prepared at different deposition temperatures under variable temperature conditions are illustrated in Figure 2a,b. These Figures, respectively, demonstrate the variations in conductivity and Seebeck coefficient of the films. A significant decrease in conductivity is observed as the temperature increases. Conversely, the Seebeck coefficient tends to increase with temperature, although the increase is relatively modest compared to the electrical conductivity. The power factor (*PF*) values of the thin film samples are evaluated in Figure 2c, with the highest value observed for the sample deposited at 200 °C. This value increases from 0.33 μWcm^−1^K^−2^ at room temperature to 1.3 μWcm^−1^K^−2^ at 300 °C. Consequently, sequent experiments were conducted at a sputtering temperature of 200 °C.

To enhance the thermoelectric properties and crystallinity of the films, the samples were annealed on a heated table at various temperatures within a glove box. Table 2 presents the compositional analysis and thickness measurements of Cu_2_Se films regulated to different annealing temperatures. A reduction in film thickness is observed concomitantly with an increase in annealing temperature, decreasing from 282 nm at 200 ℃ to 267 nm at 300 °C. Concurrently, the stoichiometric ratio of Cu to Se increases from 2.07 at 200 °C to 2.40 at 300 °C. These observations are consistent with phenomena previously reported for substrate growth temperatures, and can be primarily attributed to the enhanced volatilization of Se, facilitated by its elevated saturated vapor pressure at higher temperatures, leading to Se precipitation and loss. The XRD patterns of the thin film samples at different annealing temperatures are shown in Figure 3a. The principal diffraction peaks are discernible at 26.74° and 44.6°, corresponding to the (111) and (220) crystallographic planes of Cu_2_Se, respectively, in agreement with the standard reference pattern (PDF#06-0680). The individual XPS spectra for Cu and Se elements are shown in Figure 3b,c. Figure 3b shows the binding energy of Cu 2*p*_5/2_ and Cu 2*p*_3/2_ located at 932.5 eV and 952.3 eV. Figure 3c shows the binding energy of Se 3*d*_5/3_ and Se3*d*_2/3_ located at 53.96 eV and 54.88 eV. Comparison with standard binding energies confirms that all of the copper ions in the films exhibit a +1 valence state, while selenium ions maintain a −2 valence state. The samples retain their composition as Cu_2_Se after annealing. Figure 3d–f displays a comparative analysis of the surface morphology of the film samples across a range of annealing temperatures. The change in annealing temperature did not result in large changes in the surface morphology of the films, with similar surface crystallinity and no significant elemental enrichment.

Figure 4 shows the transmission electron microscopy (TEM) results for the films. Figure 4b,c illustrates the elemental distribution of Cu and Se in the region shown in Figure 4a, confirming a uniform distribution without noticeable elemental enrichment. Figure 4d features a high-resolution TEM image identifying a potential phase boundary in region A*, magnified in Figure 4e to better illustrate the crystalline structure of the Cu_2_Se thin film; the lattice spacing of the (111) plane measures 0.33 nm and the presumed phase boundary is clearly marked with a white dashed line in region A*. The image in region B*, which is the inverse Fourier transform of the area enclosed by the white box in Figure 4e, displays differing crystalline arrangements on either side of the phase boundary. To further explore the boundary conditions observed, a geometrical phase analysis (GPA) was conducted, with results shown in Figure 4f–i. These figures display distortion maps along the e_xx_, e_xy_, e_yx_, and e_yy_ directions, and the color bars show the induced strain/distortion in the thin film, revealing strain discrepancies due to crystal planes or grain boundaries, which may result in the scattering of phonons. Lattice defects, including point defects, facet defects, and lattice distortions, are highly susceptible to occur at the grain or phase boundaries, which are responsible for enhanced phonon scattering. Frequent phonon scattering events lead to a decrease in the mean free range of phonons, which reduces the efficiency of thermal energy transfer within the material. For semiconductors, where phonons are the main carriers of thermal energy, the increase in scattering will be the main reason for the decrease in thermal conductivity [27], so it can be extrapolated that the presence of lattice distortions in thin films will lead to an increase in phonon scattering and thus a decrease in thermal conductivity.

Figure 5 illustrates the performance variations of Cu_2_Se thin film samples prepared at different annealing temperatures. Figure 5a shows the electrical conductivity of samples prepared at different annealing temperatures as a function of temperature. As the annealing temperature increases, electrical conductivity gradually decreases. Figure 5b displays the Seebeck coefficient of the samples as a function of temperature, which shows an increasing trend with rising measurement temperature. Figure 5c depicts the power factor of the samples as a function of temperature. At room temperature, the power factor of the unannealed sample is 0.32 μWcm−^1^K−^2^, whereas the sample annealed at 300 °C exhibits a power factor of 1.43 μWcm−^1^K−^2^. With increasing test temperature, the power factor of the samples generally rises, with the sample annealed at 300 °C attaining the maximum power factor of 2.3 μWcm−^1^K−^2^ at 300 °C. Figure 5d illustrates the carrier concentration and mobility of the thin film samples quantified according to the electrical conductivity equation *σ* = *nμe*, where *n* represents the carrier concentration, *μ* denotes the mobility, and *e* is the elementary charge [28]. Increasing the annealing temperature enhances the Cu content in the samples. As the annealing temperature rises, the Cu content increases, while the carrier concentration decreases from 7.5 × 10^21^ cm−^3^ to 2.74 × 10^21^ cm−^3^. It can be inferred that the excess Cu element in the Cu_2_Se thin films mitigates the copper vacancy defects, leading to a decrease in hole concentration. This reduction in carrier concentration results in decreased scattering, thereby enhancing the carrier mobility. Consequently, electrical conductivity of the films decreases with increasing annealing temperature, whereas the Seebeck coefficient exhibits the opposite trend. However, compared to other *p*-type materials, the power factor of Cu_2_Se thin films remains relatively low, necessitating further enhancement through doping regulation.

## 4. Conclusions

We successfully fabricated pure-phase Cu_2_Se thin film samples by employing magnetron sputtering and in situ growth techniques which analyzed and compared data from XRD, XPS, and SEM. In addition, the physical mechanisms of power factor parts of thermoelectric performance enhancement were explored. Substrate heating during growth ensured the attainment of high-purity thin film samples. All of the samples manifested high carrier concentrations, hence demonstrating elevated electrical conductivity. Furthermore, annealing treatments conducted on the samples enhanced the copper content, decreasing the vacancies of Cu, thereby influencing the electrical transport properties, reducing the carrier concentration, and increasing the Seebeck coefficient and power factor of the prepared films. Consequently, the thin film prepared at a substrate temperature of 200 °C, following an annealing protocol at 300 °C for 30 min, achieved a high power factor value of 1.43 μWcm^−1^ K^−2^ at room temperature and 2.3 μWcm^−1^ K^−2^ at 300 °C.

## Figures and Tables

**Figure 1 nanomaterials-14-01421-f001:**
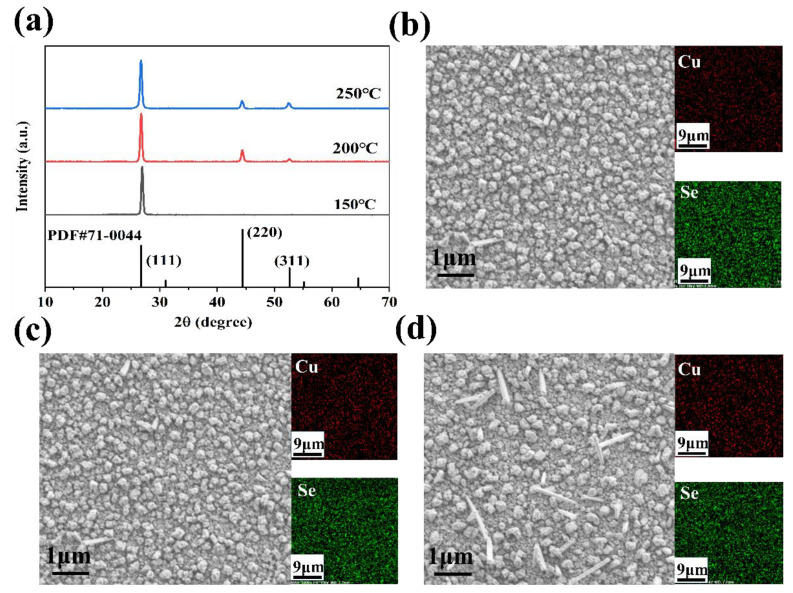
(**a**) X-ray diffraction (XRD), (**b**–**d**) SEM image of the Cu_2_Se-based thin films deposited at 150 °C, 200 °C, and 250 °C.

**Figure 2 nanomaterials-14-01421-f002:**
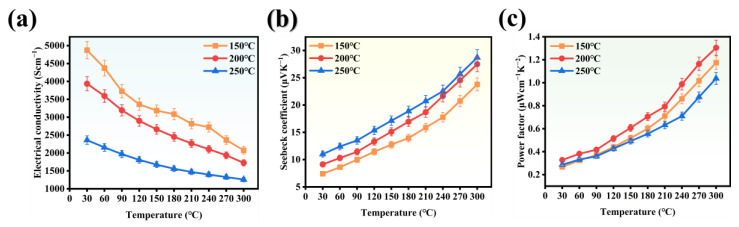
Electrical transport properties of the Cu_2_Se-based thin films grown at different substrate temperatures. (**a**) electrical conductivities *σ*, (**b**) Seebeck coefficients *S*, (**c**) power factors *PF*.

**Figure 3 nanomaterials-14-01421-f003:**
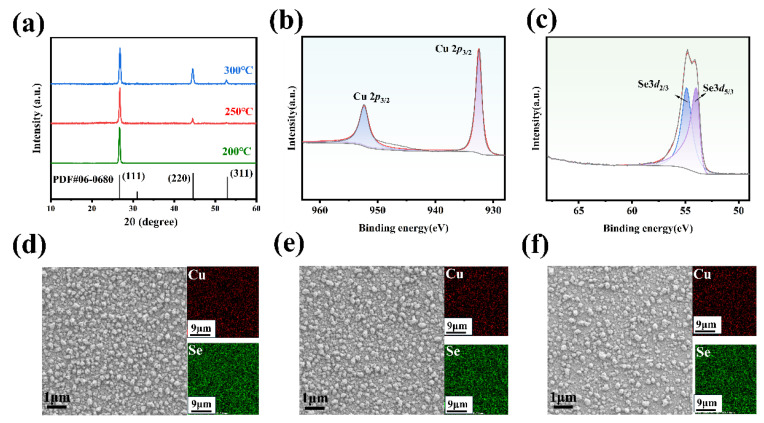
(**a**) XRD patterns of the samples at different annealing temperature, X-ray photoelectron spectra (XPS) of (**b**) Cu 2*p* and (**c**) Se 3*d*, (**d**–**f**) SEM images of Cu_2_Se-based thin films annealing at 200 °C, 250 °C, and 300 °C.

**Figure 4 nanomaterials-14-01421-f004:**
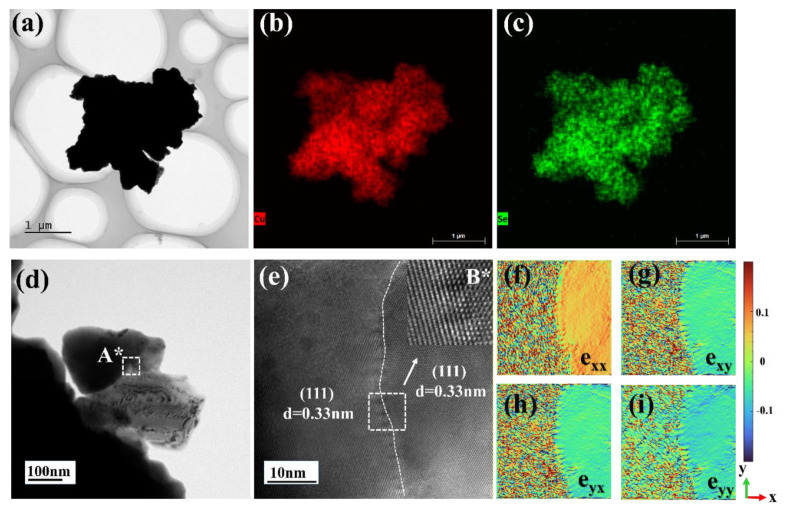
TEM analysis of the thin film annealed at 300 °C, (**a**) low-resolution TEM image; (**b**,**c**) elements mapping of (**a**); (**d**) high-resolution TEM (HRTEM) image of (**a**); (**e**) enlarged HRTEM image of selected area A* in (**d**), region B* is the inverse Fourier transform of the area enclosed by the white box in (**e**); (**f**–**i**) geometric phase analysis (GPA) strain map.

**Figure 5 nanomaterials-14-01421-f005:**
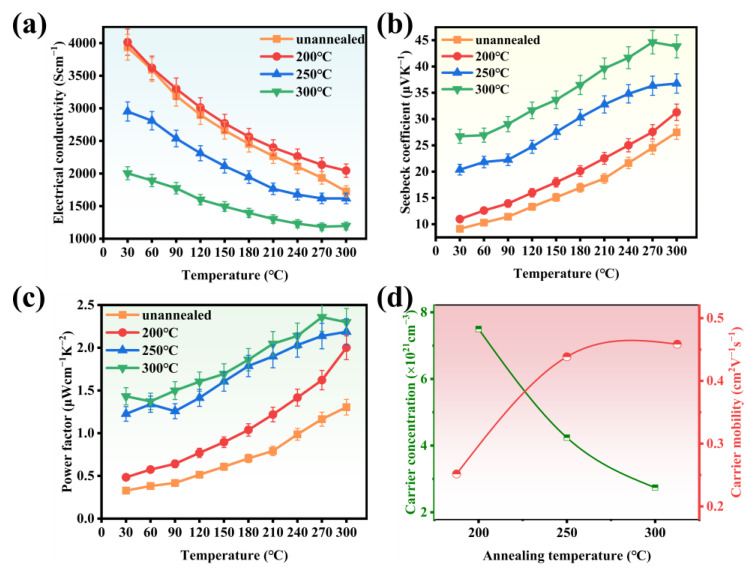
Thermoelectric properties of Cu_2_Se film prepared at different annealing temperatures. (**a**) Electrical conductivity σ, (**b**) Seebeck coefficient, (**c**) power factor *PF*, (**d**) room temperature carrier concentration and mobility.

**Table 1 nanomaterials-14-01421-t001:** Thickness and composition of the Cu_2_Se-based thin films deposited for 30 min at different substrate temperatures.

Deposition Temperature (°C)	Thickness (nm)	Cu (at.%)	Se (at.%)	Cu:Se
150	322	67.47	32.53	2.07
200	318	68.52	31.48	2.18
250	276	74.9	25.1	2.98

**Table 2 nanomaterials-14-01421-t002:** Thickness and composition of the Cu_2_Se-Based thin films annealing for 30 min at different temperatures.

Deposition Temperature (°C)	Thickness (nm)	Cu (at.%)	Se (at.%)	Cu:Se
200	282	68.73	31.27	2.2
250	245	70.01	29.99	2.33
300	267	70.56	29.44	2.40

## Data Availability

Data are contained within the article.

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
