# Peer review of "Optimization of Thermoelectric Properties and Physical Mechanisms of Cu2Se-Based Thin Films via Heat Treatment"

_nanomaterials, 2024, doi:10.3390/nano14171421_

Round 1
Reviewer 1 Report
Comments and Suggestions for Authors
In this manuscript, the authors report the TE properties of Cu2Se films deposited on flexible polyimide substrate using in-situ magnetron sputtering. The Cu2Se films deposited at 200oC exhibit good TE performance whereas annealing the samples reduces the carrier concentration and enhances the Seebeck coefficient and PF. An optimized PF of 1.43uW/cm.K2 was achieved by the authors in this study.
The following points need to be considered for possible publication in the Nanomaterials journal
1. For the annealing of samples, did the authors anneal the samples grown at 150oC at 200oC and the one grown at 200oC annealed at 250oC? Can the authors clarify? It is not clear how the authors did the annealing process.
2. In the result and discussion section: it is mentioned that there are peaks at (220) and (311) reflections present at temperatures 200oC and 250oC not for 150oC. Can the authors explain the reason for it?
3. What is the crystal structure of Cu2Se?
4.
5. Can the authors show the FWHM of (111) XRD peak and analyze its crystalline quality?
6. In Fig.1d, some rod-shaped particles are seen. Can the authors explain the reason for their formation?
7. On page 6, the authors discussed thermal conductivity from TEM analyses. How did the authors correlate strain discrepancies with thermal conductivity?
8. In the last sentence of the result and discussion on page 3: It is better to explain how columnar clusters affect the thermoelectric properties.
9. In the second line on page no: 4, it’s better to explain whether this trend is consistent across all films or if there are exceptions.
10. The paragraph on page no: 7, could include more interpretation of how these findings might impact the practical application of Cu2Se thin films in thermoelectric devices.
11. The authors should suggest some methods to improvise the PF of their films. The authors should also estimate the ZT values from the reported thermal conductivity of Cu2Se.
Comments on the Quality of English Language
The authors should revise the English of the manuscript with the help of a native English speaker.
Author Response
Comments 1: For the annealing of samples, did the authors anneal the samples grown at 150°C at 200°C and the one grown at 200°C annealed at 250°C? Can the authors clarify? It is not clear how the authors did the annealing process.
Response 1: Thanks for the reviewer’s suggestion. In our preceding exploratory work, we corroborated that 200°C represents a more optimal growth temperature by comparing the thermoelectric properties of thin films produced at room temperatures. Accordingly, the samples prepared at the 200 °C growth temperature were selected, and then subjected to heat treatment at annealing temperatures of 200 °C, 250 °C, and 300 °C, respectively. The samples were positioned with their faces upwards on a heating table within a vacuum glove box and subjected to the specified annealing temperature for a period of 30 minutes prior to their removal. We have also explained and marked “Once the samples had cooled to room temperature within the vacuum chamber, and they were subsequently transferred to a hotplate within a glove box.” (Revised manuscript, Line 107, Page 3) in the manuscript.
Comments 2: In the result and discussion section: it is mentioned that there are peaks at (220) and (311) reflections present at temperatures 200oC and 250oC not for 150oC. Can the authors explain the reason for it?
Response 2: Thank you for your suggestion, which is very meaningful. It is true that we have also found this problem you mentioned when we did the related research. At 150°C, the grains grow mainly along the (111) crystal plane, but the orientation of growth along the (220) or (331) crystal plane is not obvious. With the increase of deposition temperature, the intensity of the diffraction peaks increased significantly, and the orientation of growth along the (220) or (331) faces changed significantly. Therefore, we infer that the deposition temperature has a significant effect on the grain growth orientation in the films, which is the main reason for this phenomenon. We have also explained and marked “The crystalline grains predominantly grow along the (111) crystal plane at 150°C, rather than the (220) or (331) planes. As the deposition temperature increases, a noticeable shift in the growth orientation of the crystalline grains occurs, with enhanced prominence along the (220) and (331) planes.” (Revised manuscript, Line 135, Page 3) in the manuscript.
Comments 3: What is the crystal structure of Cu2Se?
Response 3: The crystal structure of Cu2Se exhibits variations with temperature. At room temperature, it predominantly adopts a low-temperature phase, designated as α-Cu2Se, which is characterized by a monoclinic crystal system. At elevated temperatures, this transforms into the β-Cu2Se phase, distinguished by a body-centered cubic structure. The phase transition temperature of Cu2Se is approximately 400 K. In the present experimental work, the growth temperature consistently exceeded 400 K, ensuring that all prepared samples were in the β phase. Consequently, these samples exhibited a body-centered cubic structure. We have also explained and marked “Cu2Se exhibits an α-phase at room temperature and transitions to a β-phase at elevated temperatures, with the phase transition occurring at approximately 400 K. The deposition and annealing temperatures employed in this experiment surpassed this threshold, thus ensuring that the synthesized samples were of the β-phase Cu2Se, characterized by a body-centered cubic structure.” (Revised manuscript, Line 89, Page 2) in the manuscript.
Comments 4: (No revisions were provided by the reviewers for this item.)
Comments 5: Can the authors show the FWHM of (111) XRD peak and analyze its crystalline quality?
Response 5: The software analysis of the (111) peak in the XRD spectra of the sample indicates a full width at half maximum (FWHM) of approximately 0.37721 (Figure R1). This suggests that the synthesized Cu2Se thin films predominantly grow along the (111) crystal plane. The larger grain size and high crystal quality, coupled with fewer lattice defects, contribute to the sharper diffraction peaks observed.

Figure R1. The X-ray diffraction (XRD) pattern with FWHM of (111) peak.
Comments 6: In Fig.1d, some rod-shaped particles are seen. Can the authors explain the reason for their formation?
Response 6: Thank you for your suggestions. The growth of polycrystals is influenced by various factors, including deposition temperature and deposition rate, among others. Based on the current study and speculation from the available data, the elevated energy supplied at a deposition temperature of 250°C facilitates the formation of rod- shaped particles. Future experimental work is planned to further investigate the specific physical mechanisms underlying this phenomenon.
Comments 7: On page 6, the authors discussed thermal conductivity from TEM analyses. How did the authors correlate strain discrepancies with thermal conductivity?
Response 7: Thanks to the reviewer for the meaningful suggestion. The GPA analysis of the TEM images indicates that substantial tensile strain is generated at the grain boundaries, which results in the strong scattering of phonons. Frequent phonon scattering events lead to a reduction in the mean free path of phonons, which in turn diminishes the efficiency of thermal energy transport within materials. In semiconductors and insulators, where phonons are the primary carriers of thermal energy, an increase in scattering directly results in decreased thermal conductivity. We have also explained and marked “These figures display distortion maps along the exx, exy, eyx, and eyy directions, and the color bars show the induced strain/distortion in the thin film, revealing strain discrepancies due to crystal planes or grain boundaries, which results in the scattering of phonons. Frequent phonon scattering events lead to a reduction in the mean free path of phonons, which in turn diminishes the efficiency of thermal energy transport within materials. In semiconductors and insulators, where phonons are the primary carriers of thermal energy, an increase in scattering directly results in decreased thermal conductivity .” (Revised manuscript, Line 206, Page 6) in the manuscript.
Comments 8: In the last sentence of the result and discussion on page 3: It is better to explain how columnar clusters affect the thermoelectric properties.
Response 8: Combined with the observed trends in the thermoelectric properties shown in Figure 2, it is noted that films containing rod-shaped particles exhibit a significant decrease in electrical conductivity and an increase in the Seebeck coefficient. It is initially hypothesized that the presence of rod-shaped particles enhances the number of interfaces, thereby facilitating the phenomenon of energy filtering. This effect selectively filters out lower-energy carriers, leading to an increase in the Seebeck coefficient. This conclusion represents a preliminary inference based on the current experimental data. Further investigations into the underlying mechanisms of this phenomenon will be conducted in subsequent studies.
Comments 9: In the second line on page no: 4, it’s better to explain whether this trend is consistent across all films or if there are exceptions.
Response 9: The trends related to the sample behavior with temperature variations are consistent throughout the text. As the test temperature increases, there is a noted decrease in the electrical conductivity of the samples, while the Seebeck coefficient exhibits an upward trend. Consequently, the overall power factor value increases, without any exceptions to these observed trends.
Comments 10: The paragraph on page no: 7, could include more interpretation of how these findings might impact the practical application of Cu2Se thin films in thermoelectric devices.
Response 10: Thanks to the reviewer for the meaningful suggestion. High thermoelectric property in thin film is determined by high Seebeck coefficient, high electrical conductivity, and low thermal conductivity. We have successfully fabricated thin films exhibiting these properties. In subsequent work, we will further enhance the thermoelectric properties by doping elements, laying a solid foundation for the future fabrication of devices such as micro-thermoelectric power generating sensors, thermoelectric generators, and wearable flexible devices.
Comments 11: The authors should suggest some methods to improvise the PF of their films. The authors should also estimate the ZT values from the reported thermal conductivity of Cu2Se.
Response 11: Thanks to the reviewer for the meaningful suggestion. In the subsequent experimental work, we plan to modulate the carrier concentration in the sample by doping other elements such as Ag, Al, etc. to improve the Seebeck coefficient of the sample. As the experimental work is still in the preliminary exploration phase and the thermal conductivity measurement of the thin films is subject to significant errors, actual measurements of thermal conductivity were not conducted. According to the references [1], the thermal conductivity of bulk Cu2Se is approximately 0.4 Wm-1K-1. Using the thermoelectric figure of merit, ZT = S2σT/κ, the ZT value for the thin film sample is estimated to be approximately 0.33. Since the thermal conductivity of the sample was not actually measured, this data will not be included in the main text of the article.
Reviewer 2 Report
Comments and Suggestions for Authors
REVIEW
Key reasons to show the significance of work to capture the reader's interest.
1. The abstract should describe "thermoelectric properties" due to their significance in all scientific research.
2. In the introduction, it is important to describe the significance of each thermoelectric property variable and provide supporting references.
3. Currently, most thermoelectric films are fabricated using Bi- and Sb- based materials, which exhibit high thermoelectric properties at room temperature.
4. The authors should include relevant references for the statement, "Currently, most thermoelectric films are fabricated using Bi- and Sb- based materials, which exhibit high thermoelectric properties at room temperature." Lines 42-43.
5. Materials and Methods: Regarding the thermal treatment of "annealed temperature," could you please provide experimental details such as the heating speed, presence of gas, and volume?
6. Make clear whether the heat treatment was in situ or ex-situ in the body of the paper.
7. In the conclusions section concerning the physical term "thermoelectric properties," it is necessary to include them. Other references in the literature are cited.
8. The phrase "physical mechanisms" is only in the title, so it must be added in the introduction with its references and in the results and conclusions sections.
9. The phrase "physical mechanisms" only appears in the title. Thus, it should be included in the introduction, references, and the results and conclusions sections.
10. The "heat treatment" cited very few sources, indicating the need for more comprehensive citations given the importance of heat treatment.
11. They conducted X-ray diffraction measurements but did not describe the results in the abstract and conclusions. Additionally, they failed to include references from the cited literature. This raises the question: Are no scientific works available on this subject?
12. In the abstract and conclusions, it's important for the authors to include an analysis of the "SEM image" measurements to emphasize their results. Additionally, it's unclear whether the literature was referenced or if the work is entirely original. Can you please explain further?
13. To captivate readers' interest, it is important to include significant comments from the abstract, introduction, and conclusions on "Electrical transport properties" and “electrical conductivities” along with relevant literature.
14. The abstract and conclusions should include the relevant results of the "XPS" measurements.
Comments on the Quality of English LanguageWith the requested corrections, the English needs to be reviewed
Author Response
Comments 1: The abstract should describe "thermoelectric properties" due to their significance in all scientific research.
Response 1: Thanks to the reviewer for the meaningful suggestion. Thermoelectric properties refer to the characteristics exhibited by materials under the influence of thermoelectric effects. These properties of thermoelectric materials are typically described using several key parameters. The thermoelectric performance of thin films can be quantified by the dimensionless figure of merit, ZT, which is defined as ZT=S2σT/κ, where S is the Seebeck coefficient, σ is the electrical conductivity, T is the absolute temperature, and κ is the thermal conductivity. It is evident that high Seebeck coefficient, high electrical conductivity and low thermal conductivity are essential characteristics for achieving efficient thermoelectric conversion Due to the challenges associated with measuring κ in thin films, the power factor PF = S2σ is commonly employed to evaluate thermoelectric properties. We have also explained and marked “The efficiency of thermoelectric materials can be assessed by the figure of merit ZT (ZT = S2σT∕κ, where S, σ, κ, and T represent the Seebeck coefficient, electrical conductivity, total thermal conductivity and absolute temperature, respectively). Due to the challenges associated with measuring κ in thin films, the power factor PF = S2σ is commonly employed to evaluate thermoelectric properties.” (Revised manuscript, Line 40, Page 1) in the manuscript. To ensure concise and streamlined language within the article, we have generalized the text by collectively referring to ZT values, PF values, and other physical parameters as "thermoelectric properties."
Comments 2: In the introduction, it is important to describe the significance of each thermoelectric property variable and provide supporting references.
Response 2: Thank you for your suggestions. These properties of thermoelectric materials are typically described using several key parameters. The thermoelectric performance of thin films can be quantified by the dimensionless figure of merit, ZT, which is defined as ZT=S2σT/κ, where S is the Seebeck coefficient, σ is the electrical conductivity, T is the absolute temperature, and κ is the thermal conductivity. It is evident that high Seebeck coefficient, high electrical conductivity and low thermal conductivity are essential characteristics for achieving efficient thermoelectric conversion. Due to the challenges associated with measuring κ in thin films, the power factor PF = S2σ is commonly employed to evaluate thermoelectric properties. We have also explained and marked “The efficiency of thermoelectric materials can be assessed by the figure of merit ZT (ZT = S2σT∕κ, where S, σ, κ, and T represent the Seebeck coefficient, electrical conductivity, total thermal conductivity and absolute temperature, respectively). Due to the challenges associated with measuring κ in thin films, the power factor PF = S2σ is commonly employed to evaluate thermoelectric properties.” (Revised manuscript, Line 40, Page 1) in the manuscript.
Comments 3: Currently, most thermoelectric films are fabricated using Bi- and Sb- based materials, which exhibit high thermoelectric properties at room temperature.
Response 3:Thanks to the reviewer for the meaningful suggestion. Common thermoelectric materials currently include Bi-Te and Sb-Te based compounds. However, due to the limited availability of tellurium and its slight toxicity, there is a need to explore alternative materials to reduce costs and enhance environmental sustainability. We have also explained and marked “Currently, most thermoelectric films are fabricated using Bi-Te and Sb-Te based materials, which exhibit high thermoelectric properties at room temperature. However, due to their toxicity and high-cost implications they pose difficulties for industrial-scale production. Therefore, there is a need to identify an environmentally friendly material with low toxicity and cost-effectiveness that possesses a high ZT value for preparing thermoelectric thin films.” (Revised manuscript, Line 42, Page 1) in the manuscript.
Comments 4: The authors should include relevant references for the statement, "Currently, most thermoelectric films are fabricated using Bi- and Sb- based materials, which exhibit high thermoelectric properties at room temperature." Lines 42-43.
Response 4: Thanks to the reviewer for the meaningful suggestion. Common thermoelectric materials currently employed are Bi-Te and Sb-Te based compounds. Nevertheless, due to the limited availability of tellurium and its mild toxicity, it is imperative to investigate alternative materials that could lower costs and increase environmental sustainability. We have also explained and marked “Currently, most thermoelectric films are fabricated using Bi-Te and Sb-Te based materials, which exhibit high thermoelectric properties at room temperature. However, due to their toxicity and high-cost implications they pose difficulties for industrial-scale production. Therefore, there is a need to identify an environmentally friendly material with low toxicity and cost-effectiveness that possesses a high ZT value for preparing thermoelectric thin films.” (Revised manuscript, Line 42, Page 1) in the manuscript.
Comments 5: Materials and Methods: Regarding the thermal treatment of "annealed temperature," could you please provide experimental details such as the heating speed, presence of gas, and volume?
Response 5: Thanks to the reviewer for the meaningful suggestion. The hotplate requires 15 minutes to heat from 25°C to 300°C, with an average temperature increase of approximately 18.33°C per minute. The glove box is filled with a mixture of nitrogen and oxygen gases, and the internal pressure during heating is maintained at 0.31 mbar.
Comments 6: Make clear whether the heat treatment was in situ or ex-situ in the body of the paper.
Response 6:Thank you for your suggestions. The heat treatment was ex-situ. Following the completion of the thin film fabrication, the sample was allowed to naturally cool to room temperature within the magnetron sputtering chamber before being removed. Subsequently, the sample was transferred to a heating stage within a glove box for thermal treatment. We have also explained and marked “Once the samples had cooled to room temperature within the vacuum chamber, and they were subsequently transferred to a hotplate within a glove box.” (Revised manuscript, Line 107, Page 3) in the manuscript.
Comments 7: In the conclusions section concerning the physical term "thermoelectric properties," it is necessary to include them. Other references in the literature are cited.
Response 7: Thank you for your suggestions. These properties of thermoelectric materials are typically described using several key parameters. The thermoelectric performance of thin films can be quantified by the dimensionless figure of merit, ZT, which is defined as ZT=S2σT/κ, where S is the Seebeck coefficient, σ is the electrical conductivity, T is the absolute temperature, and κ is the thermal conductivity. It is evident that high Seebeck coefficient, high electrical conductivity and low thermal conductivity are essential characteristics for achieving efficient thermoelectric conversion Due to the challenges associated with measuring κ in thin films, the power factor PF = S2σ is commonly employed to evaluate thermoelectric properties. We have also explained and marked “The efficiency of thermoelectric materials can be assessed by the figure of merit ZT (ZT = S2σT∕κ, where S, σ, κ, and T represent the Seebeck coefficient, electrical conductivity, total thermal conductivity and absolute temperature, respectively). Due to the challenges associated with measuring κ in thin films, the power factor PF = S2σ is commonly employed to evaluate thermoelectric properties.” (Revised manuscript, Line 40, Page 1) in the manuscript.
Comments 8: The phrase "physical mechanisms" is only in the title, so it must be added in the introduction with its references and in the results and conclusions sections.
Response 8: Thank you for your suggestions. The physical mechanisms referred to in this article pertain to the reasons behind changes in the thermoelectric properties of thin films and the fundamental physical nature causing these changes. The investigation into the thermoelectric performance variations through thermal treatment, along with discussions on the corresponding microstructural tests, elucidates these physical essentials. Specifically, in the text, the physical essence is manifested through changes in the carrier concentration and mobility within the films. We have also marked “This study focuses on preparing Cu2Se flexible thin films through in-situ magnetron sputtering technology while carefully optimizing key preparation parameters and explored the physical mechanism of thermoelectric property enhancement.” (Revised manuscript, Line 18, Page 1) in the manuscript and “In addition the physical mechanisms of thermoelectric performance enhancement were explored.” (Revised manuscript, Line 246, Page 8) in the manuscript.
Comments 9: The phrase "physical mechanisms" only appears in the title. Thus, it should be included in the introduction, references, and the results and conclusions sections.
Response 9: Thank you for your suggestions. The physical mechanisms discussed in this article relate to the underlying causes of variations in the thermoelectric properties of thin films and the fundamental physical principles driving these changes. This study examines the fluctuations in thermoelectric performance through thermal treatments and explores their correlation with microstructural alterations. The text specifically attributes these physical fundamentals to changes in the carrier concentration and mobility within the films. We have also marked “This study focuses on preparing Cu2Se flexible thin films through in-situ magnetron sputtering technology while carefully optimizing key preparation parameters and explored the physical mechanism of thermoelectric property enhancement.” (Revised manuscript, Line 18, Page 1) in the manuscript and “In addition the physical mechanisms of thermoelectric performance enhancement were explored.” (Revised manuscript, Line 246, Page 8) in the manuscript.
Comments 10: The "heat treatment" cited very few sources, indicating the need for more comprehensive citations given the importance of heat treatment.
Response 10: Thanks to the reviewer for the meaningful suggestion.Heat treatment is a critical step in the preparation and performance optimization of thin films and is widely employed as a conventional method in experimental work. Due to the common and fundamental heat treatment methods used in the experimental work, extensive references were not cited.
Comments 11: They conducted X-ray diffraction measurements but did not describe the results in the abstract and conclusions. Additionally, they failed to include references from the cited literature. This raises the question: Are no scientific works available on this subject?
Response 11: Thank you for your suggestions. X-ray diffraction (XRD) is a fundamental analytical technique whose principles, results, and analyses are widely recognized in numerous studies. The PDF card data utilized originates from an internationally authoritative database, hence specific citations are not provided. Detailed analysis and discussion of diffraction peaks and orientation information are presented on page 3, line 131, and page 5, line 171 of the main text. Additionally, an analysis related to XRD is also incorporated in the conclusions section of the document. We have marked “We successfully fabricated pure-phase Cu2Se thin film samples by employing magnetron sputtering and in-situ growth techniques, which analyzed and compared data from XRD, XPS, and SEM.” (Revised manuscript, Line 244, Page 8) in the manuscript
Comments 12: In the abstract and conclusions, it's important for the authors to include an analysis of the "SEM image" measurements to emphasize their results. Additionally, it's unclear whether the literature was referenced or if the work is entirely original. Can you please explain further?
Response 12: Thanks to the reviewer for the meaningful suggestion. The SEM image analysis and test results are original. As SEM images reflect the conditions of different samples, it is not feasible to make identical reference. By analyzing the XRD and SEM test results, we successfully synthesized high-purity Cu2Se, which also validates the appropriateness of the annealing and growth temperatures selected during the experimental work.
Comments 13: To captivate readers' interest, it is important to include significant comments from the abstract, introduction, and conclusions on "Electrical transport properties" and “electrical conductivities” along with relevant literature.
Response 13: Thank you for your suggestion. For the purposes of this paper, electrical transport properties are the broad term that refers to the electrical conductivity of a material, including resistance, conductivity, Hall effect, etc. In thermoelectric materials we generally study the transport properties of their carriers, and the electrical transport properties are mainly studied in terms of the changes in carrier concentration and mobility in the thin film samples, which are ultimately manifested as changes in conductivity. We have described and analyzed the carrier concentration, mobility and conductivity in the corresponding parts of the Abstract, Introduction, and Conclusion, especially in the Introduction, and also made relevant references, such as reference [22]. The analysis of carrier-related data in this paper is original, so no reference is cited in the data analysis part.
Comments 14: The abstract and conclusions should include the relevant results of the "XPS" measurements.
Response 14: The analysis of XPS data confirms that the synthesized sample is pure-phase Cu2Se, with no other chemical valence states present. After some discussion among our authors, we agreed that in order to keep the abstract concise and focused, we would no longer add this section to the abstract, but we added it to the conclusion part. We marked “We successfully fabricated pure-phase Cu2Se thin film samples by employing magnetron sputtering and in-situ growth techniques, which analyzed and compared data from XRD, XPS, and SEM.” (Revised manuscript, Line 244, Page 8) in the manuscript